# Hit screening with multivariate robust outlier detection

Hui Sun Leong[1]*, Tianhui Zhang[2], Adam Corrigan[1], Alessia Serrano[3], Ulrike Künzel[3], Niamh Mullooly[3], Ceri Wiggins[3], Yinhai Wang[1], Steven Novick[2]

1 Data Sciences and Quantitative Biology, Discovery Sciences, Biopharmaceuticals R&D, AstraZeneca, Cambridge, United Kingdom, 2 Data Sciences and Quantitative Biology, Discovery Sciences, Biopharmaceuticals R&D, AstraZeneca, Gaithersburg, Maryland, United States of America, 3 Functional Genomics, Discovery Biology, Discovery Sciences, Biopharmaceuticals R&D, AstraZeneca, Cambridge, United Kingdom

◎ These authors contributed equally to this work.

* huisun.leong@astrazeneca.com

**Data Availability Statement:** All relevant data are within the manuscript and its Supporting Information files.

**Funding:** The author(s) received no specific funding for this work.

## Abstract

Hit screening, which involves the identification of compounds or targets capable of modulating disease-relevant processes, is an important step in drug discovery. Some assays, such as image-based high-content screenings, produce complex multivariate readouts. To fully exploit the richness of such data, advanced analytical methods that go beyond the conventional univariate approaches should be employed. In this work, we tackle the problem of hit identification in multivariate assays. As with univariate assays, a hit from a multivariate assay can be defined as a candidate that yields an assay value sufficiently far away in distance from the mean or central value of inactives. Viewed another way, a hit is an outlier from the distribution of inactives. A method was developed for identifying multivariate hit in high-dimensional data sets based on principal components and robust Mahalanobis distance (the multivariate analogue to the $Z$- or $T$-statistic). The proposed method, termed mROUT (multivariate robust outlier detection), demonstrates superior performance over other techniques in the literature in terms of maintaining Type I error, false discovery rate and true discovery rate in simulation studies. The performance of mROUT is also illustrated on a CRISPR knockout data set from in-house phenotypic screening programme.

## Introduction

Hit screening is an integral component of the preclinical workflow, from identification and validation of druggable targets through to the identification of lead compounds. Single-shot hit identification is traditionally performed with a univariate assay by comparing the assay signal of a compound or other external perturbation (hereafter, compound) to that of a set of inactive compounds or to one or more controls [1]. While there are many procedures to detect activity in a hit screening assay, this work will focus on techniques for which activity of a compound is determined by calculating the distance between the assay signal of a single compound to the mean signal of inactives. If the distribution of inactives is assumed to be normal (a.k.a.,

**Competing interests:** The authors have declared that no competing interests exist.

*Gaussian*) and the assay output is univariate, then distance may be calculated as the number of standard deviations between the two. Some hit screening assays, such as Cell Painting [2] or other morphological profiling, produce multivariate output that cannot be reduced to a single dimension. In such a case, a multivariate metric called Mahalanobis distance [3] is calculated to measure the gap between a compound and the mean of inactives. Whether the data are univariate or multivariate, hit screening may be recast as an outlier detection exercise, i.e., the compound of interest is declared to be active if it falls outside of the normal population distribution of the inactive compounds. For the $i^{\text{th}}$ compound, the two competing hypotheses of interest are

$$H_0 : \text{ The } i^{\text{th}} \text{ compound belongs to the normal distribution of inactives,}$$

$$H_1 : \text{ The } i^{\text{th}} \text{ compound does not belong to the normal distribution of inactives.} \tag{1}$$

In this work, a method for evaluating hypotheses in (1) by detecting outliers for multivariate normal data is proposed and evaluated for its properties of Type I error, false discovery rate, and true discovery rate. The method is demonstrated via simulation as well as on a real 96-dimensional CRISPR [4] gene knockout hit screening bioassay. To initiate the reader, it may be valuable to review univariate systems for hit screening and outlier detection.

## Univariate outlier detection

For notation purposes, let $X_i$ denote the assay value of the $i^{\text{th}}$ compound ($i = 1, \ldots, N$), $\bar{X} = \frac{1}{N}\sum_i X_i$ denote the sample mean, and $\sigma$ denote the population standard deviation of inactives. One of the simplest systems to detect activity is given by Makarenkov et al. (Method 1 in [5]) who identify the $i^{\text{th}}$ compound as active if $|X_i - \bar{X}| > c \times \sigma$, where $c$ is an appropriately preselected constant. Although $\sigma$ cannot be known exactly, with enough experience with an assay, the value of $\sigma$ may be determined with superb precision. The Makarenkov rule with $c = 3$ would correctly identify the outliers in Fig 1, which displays computer-generated data from a normal distribution with true population mean $\mu = 10$ and standard deviation $\sigma = 3$ for a set of 25 inactive compounds (circles) with three outliers (triangles) that rise more than three standard deviations above the distribution. Through simple algebra, the value $c$ may also be seen as the lower threshold for a $Z$-score so that a compound is declared active (i.e., $H_1$ is declared in (1)) if $Z_i = \frac{|X_i - \bar{X}|}{\sigma} > c$.

Another procedure for selecting actives is given by the Minimum Discriminatory Difference (MDD) [6,7], which discriminates the signal between two compounds if the distance between their respective measurements is larger than $\Phi^{-1}(0.975)\sqrt{2}\sigma$, where $\Phi^{-1}(.)$ is the inverse of the cumulative normal probability density function (pdf). The MDD statistic can be modified to compare $X_i$ to $\bar{X}$, declaring a compound as active if $|X_i - \bar{X}| > \left\{\Phi^{-1}\left(0.975\right)\sqrt{1 - \frac{1}{N}}\right\}\sigma$ or, with some algebraic manipulation, if $Z_{i,MDD} = \frac{|X_i - \bar{X}|}{\sigma\sqrt{1 - \frac{1}{N}}} > \Phi^{-1}(0.975)$. In many situations, the value of $\sigma$ is not known and is estimated by the sample standard deviation $s$ with $\lambda$ degrees of freedom. In that case, a $T$-score and rule to declare $H_1$ may be created by $T_{i,MDD} = \frac{|X_i - \bar{X}|}{s\sqrt{1 - \frac{1}{N}}} > t^{-1}(0.975, \lambda)$, where $t^{-1}(.,\lambda)$ is the inverse central $T$ pdf with $\lambda$ degrees of freedom.

The formulation of MDD assumes that the calculation of sample mean $\bar{X}$ includes the $i^{\text{th}}$ compound assay value $X_i$. In the situation for which $X_i$ is not included in $\bar{X}$, then the standard

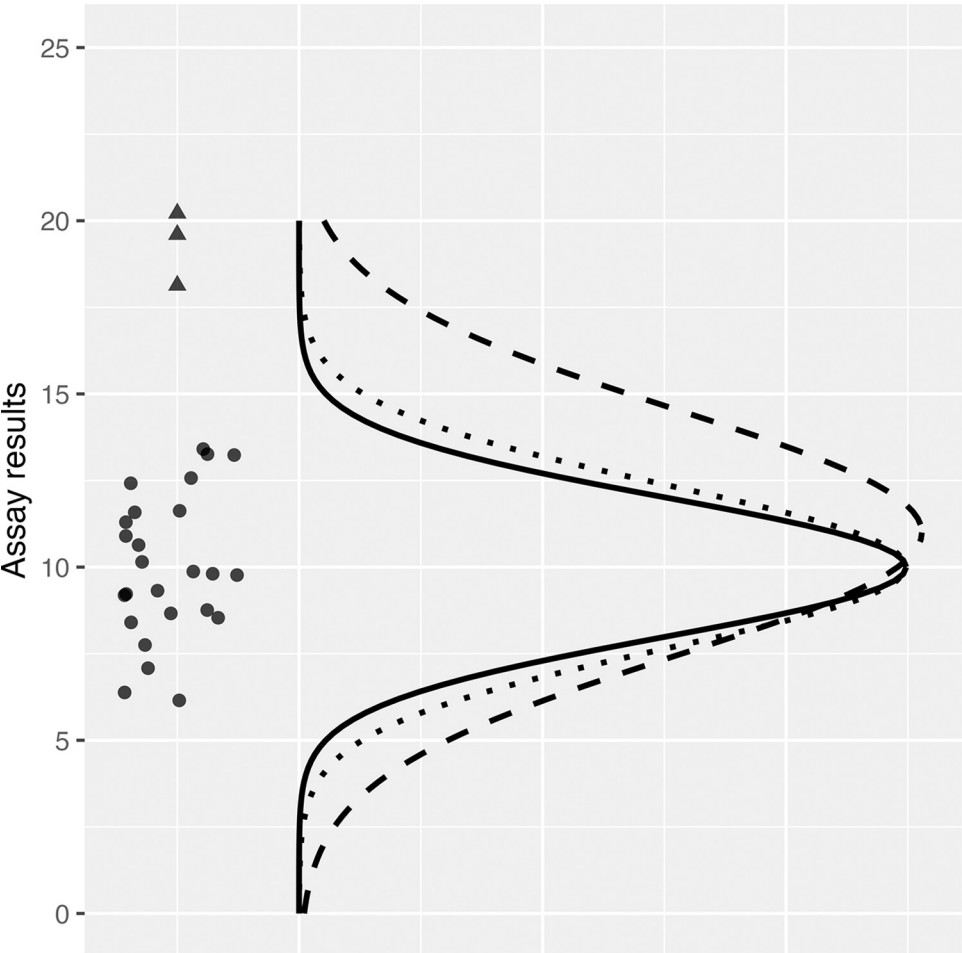

**Fig 1. Sample mean and standard deviation estimates are influenced by the presence of outliers.** The graph shows observations from a normal distribution (circles) with mean = 10 and standard deviation = 3 and outliers (triangles). Solid, dashed, and dotted lines show the true population distribution, the normal distribution based on sample mean and standard deviation, and the normal distribution based on the median and median absolute deviation (MAD), respectively.

error of $X_i - \bar{X}$ is $\sigma\sqrt{1 + \frac{1}{N}}$. In this case, the *T*-score version of MDD is equivalent to a prediction interval outlier detection method in the pharmaceutical manufacturing literature [8,9], which declares $X_i$ as an outlier if $|X_i - \bar{X}| > \left\{ t^{-1}(0.975, \lambda)\sqrt{1 + \frac{1}{N}} \right\} s$ or $T_{i,pi} = \frac{|X_i - \bar{X}|}{s\sqrt{1 + \frac{1}{N}}} > t^{-1}(0.975, \lambda)$. For large values of *N*, the difference between $T_{i,MDD}$ and $T_{i,pi}$ is negligible. By removing *N* completely, $T_{i,sr} = \frac{|X_i - \bar{X}|}{s} > t^{-1}(0.975, \lambda)$ becomes an outlier rule for standardized residuals.

In the presence of outliers, it is well known that the sample mean and standard deviation estimates can be biased, which can lead to false negatives (outliers are not detected) as well as false positives (inliers are detected). The problem of bias is well-demonstrated by the data shown in Fig 1, where the population mean is 10 and the standard deviation is 3 for inactives. The sample mean and standard deviation of the inactives (circles) are very close to their true values, but the overall sample mean (including outliers) is 11.5 (biased by 1.5 units) and the overall sample standard deviation is 5.3 (1.8 times inflation). Bias in the center and spread can

result in falsely declaring $H_0$ (false negatives) and/or falsely declaring $H_1$ (false positives) in (1). These effects are sometimes respectively called masking and swamping [10]. To mitigate these problems, so long as the percentage of outliers is small enough, various authors [11–13] propose robust measures of center and spread that are little influenced by the outliers in the system, such as the median and MAD. Applying robust statistics to the data in Fig 1, the median and MAD of the full data are 10.0 and 3.5, which are closer to the population values. Hampel's outlier detection rule [14], which is a robust version of Method 1 from Makarenkov et al. [5], substitutes the median and MAD for the mean and standard deviation. Brideau et al. [11] propose the B ("better") score, which is similar to Hampel's rule, but also robustly models the row and column effects from an experimental plate using median polishing. Other robust methods to calculate the center and spread of a normal distribution with outliers include trimming, winsorizing, and weighting. Sondag et al. [12] propose a robust version of the prediction-interval method ($T_{i,pi}$) that uses iterative weighting. For references on robust statistical methods, see [13,15,16].

Another robust method for outlier-contaminated normal data uses a symmetric, wide-tailed model, such as the Cauchy distribution, to locate the center of the inlier distribution. Motulsky and Brown [17] follow this track and propose *robust outlier detection* (ROUT) to estimate the mean of a normal distribution with a Cauchy location statistic ($\tilde{\mu}$). The authors estimate the standard deviation with the robust standard deviation of residuals (RSDR), where RSDR is the 68.27th percentile of the absolute residuals $|X_i - \tilde{\mu}|$. Sondag et al. [12] demonstrate that ROUT controls the false positive rate while providing an acceptable true positive rate. An outlier is declared by ROUT if the Benjamini-Hochberg [18] (hereafter, BH) multiplicity-adjusted $p$-value is less than a pre-determined false discovery rate $Q$ (Motulsky and Brown suggest using $Q = 0.01$), where, under $H_0$ in (1), the $p$-value is $2 \times \Pr\left(T > \frac{|X_i - \tilde{\mu}|}{RSDR}\right)$ and $T$ is central t-distributed with $N$—1 degrees of freedom.

## Multivariate outlier detection

Much of the literature for multivariate hit screening follows a similar logic by identifying outlier compounds that are a detectable distance from a robust estimate of the multidimensional mean of inactives. Statistical methods of multivariate outlier detection compute the distance of each observation from the center of the data and classify those above a certain threshold as outliers. Let $W \sim MVN(\mu, \Sigma)$ denote a $p$-dimensional vector. The conventional measure of distance between $W$ and $\mu$ is the Mahalanobis distance given by $MD_i = \sqrt{(W - \mu)' \Sigma^{-1} (W - \mu)}$. When $\mu$ and $\Sigma$ are known, it is straight-forward to show that the squared Mahalanobis distance follows a chi-square ($\chi^2$) distribution with $p$ degrees of freedom, $MD_i^2 \sim \chi^2(p)$.

For hit screening, let $X_i$ and $\bar{X}$ denote the $p$-dimensional vectors for the $i$th compound and the mean across compounds. The Mahalanobis distance analogue of the univariate MDD $T$-score is

$$MD_i = \sqrt{(X_i - \bar{X})' \left\{ \left(1 - \frac{1}{N}\right) \hat{\Sigma} \right\}^{-1} (X_i - \bar{X})}, \tag{2}$$

where $\hat{\Sigma}$ is the sample variance-covariance estimator for $\Sigma$. Note that $MD_i^2$ from Eq (2) is also called a Hotelling's $T$-squared statistic [19], which tests if the mean of $X_i$ is equal (or not) to the mean of $\bar{X}$. Under $H_0$ in (1), $\frac{N-p}{(N-1)p} MD_i^2 \sim F(p, N-p)$ distributed. It follows that the $p$-value to test the hypotheses in (1) is given by $\Pr\left(F^* > \frac{N-p}{(N-1)p} MD_i^2\right)$, where $F^* \sim F(p, N-p)$. As in the

univariate case, for the situation in which $X_i$ is not part of the calculation of $\bar{X}$, then the variance-covariance term in Eq (2) is $\left(1 + \frac{1}{N}\right)\hat{\boldsymbol{\Sigma}}$.

Because masking and swamping can occur in the multivariate space, it is often desirable to compute a robust version of the Mahalanobis distance using robust replacements of $\bar{X}$ and $\hat{\boldsymbol{\Sigma}}$ (say, $\tilde{\boldsymbol{\mu}}$ and $\tilde{\boldsymbol{\Sigma}}$) that are less affected by outlying observations in Eq (2). Many robust procedures to calculate $\tilde{\boldsymbol{\mu}}$ and $\tilde{\boldsymbol{\Sigma}}$ have been proposed in the literature, including the minimum covariance determinant (MCD) estimator [20–22], S-estimators [23], MM-estimators [24], the orthogonalized Gnanadesikan-Kettenring (OGK) estimator [25], and shrinkage procedures [26]. When used to calculate Eq (2), these methods are called robust Mahalanobis distance (rMD) estimators.

Although the MCD estimator of Rousseeuw is widely used for rMD, Adrover and Yohai [27] report that MCD becomes unreliable as $p$ becomes large. Other rMD methods may fare better, but because the number of parameters increases quadratically with $p$, it is likely other robust procedures for estimating $\boldsymbol{\mu}$ and $\Sigma$ show a similar reliability issue. For large $p$, a dimensionality reduction step via principal components analysis (PCA) can be performed. Hubert et al. [28] claims that standard PCA may be unreliable if the data contain outliers and suggests several robust PCA (rPCA) algorithms. Various authors [28–32], evaluate (1) by implementing a rPCA step, followed by a rMD testing procedure on the reduced-dimension scores matrix. See Todorov and Filzmoser [32] for an overview of this procedure. Unlike classic PCA (cPCA), the rMD scores columns are not necessarily orthogonal so the variance-covariance matrix in Eq (2) must be estimated; however, because of the reduced dimensionality in the rPCA scores space, the rMD reliability may be less of an issue. More modern dimensionality-reduction techniques, such as t-distributed Stochastic Neighbor Embedding (t-SNE) [33], Uniform Manifold Approximation and Projection (UMAP) [34], Independent Component Analysis (ICA) [35], kernel PCA [36], and Local Linear Embedding with Adaptive Neighbors (LLEAN) [37], might also be considered and may provide better separation in the data. However, it is unlikely that these methods would produce scores that are multivariate normally distributed, meaning that alternative approaches to the F-statistic would need to be considered.

The aforementioned authors who base outlier detection of Mahalanobis distance in a robust PCA space do not calculate a $p$-value or other probability metric, but instead provide a flag to indicate whether an observation is declared an outlier. This operation can result in uncontrolled false discovery rate of outliers, which is a major drawback. By contrast, the method proposed in this work, which performs cPCA followed by rMD, tests the hypotheses in (1) with a BH FDR-adjusted $p$-value. By way of explanation for the cPCA step, our method requires orthogonality of the scores matrix. To enable a direct comparison of our method and the rPCA-rMD approaches, we adjusted the latter from returning an outlier flag to returning a $p$-value based on the distance metrics and distributional assumption (chi-square) proposed by the authors, and further corrected the $p$-value for multiple testing using the BH FDR correction method. That is, the authors provided an outlier flag if $\hat{d} > \chi^2(r)$, where $\chi^2$ is their test statistic and $r$ is their calculated degrees of freedom. Even after we modified rPCA-rMD methods to also provide a BH FDR-adjusted $p$-value and despite the warning of using cPCA by Hubert et al. [28], we found that our method had better Type I error control, better FDR control, and/ or higher statistical power to declare $H_1$ in (1) compared with other methods. The performance of the proposed approach was evaluated through simulation studies with $p = 2, 3$, and 96, where the case $p = 96$ mimics a real data situation. The proposed method was also applied to a real CRISPR data set from in-house hit screening programme.

While beyond the scope of this work, it is worth considering other advanced multivariate outlier detection methods that are capable of handling nonlinear relationship, non-stationary

data and large-scale datasets, such as [38–40]. These methods typically involve the conceptualization of at least two groups, namely inactives versus others, through classification or clustering. While these techniques may offer advantages in some scenarios, our proposed method focuses solely on determining whether an observation belongs to the inactives group, eliminating the need for assumption about the data being separable into more than one group, therefore simplifies the analysis and enhances efficiency.

## Materials and methods

### The mROUT algorithm

Motivated by the robust outlier detection method (ROUT) of Motulsky and Brown [17], we propose a method called multivariate ROUT (mROUT) for identifying outliers in high-dimensional data using a cPCA-rMD approach. The procedure mROUT borrows from Filzmoser, Maronna, and Werner [31] and Motulsky and Brown [17], and consists of the following broad steps.

1. Dimensionality reduction

2. Robust estimation of location and scatter

3. Outlier detection via Hotelling's $T$-squared testing

It is assumed that the majority of compounds (rows of the $N \times p$ matrix $X$) will show no activity. Further, assay data for inactive compounds are assumed to follow a multivariate normal distribution, i.e., $X_i \sim MVN(\boldsymbol{\mu}, \Sigma)$. No distribution assumption is placed on active compounds, which, under $H_1$ of (1) is not part of the multivariate normal distribution.

The assumption that inactive compounds follow a multivariate normally distribution is not likely universally valid. Later, we describe processing of images using a multi-task convolutional neural network (CNN) approach. For our real in-house data sets, the CNN appears to produce Gaussian-like data columns. For profiles generated with other image analysis software, the data may exhibit skewed or other non-normal characteristics. The recommended practice by Caidedo et al. [41] is to transform feature values with mathematical operations, such that the feature values approximate a normal distribution, mean centered and have comparable standard deviation to facilitate downstream analysis. Therefore, we think our proposed approach will be generally applicable for many cases after appropriate transformation or preprocessing of image features.

The essence of our work is that we are creating a robust distribution (after an appropriate dimensionality reduction) for inactives and then asking whether each observation belongs to the distribution or not. Big deviations from the Gaussian distribution will require a different robust distribution for the inactives and different test statistic. Such a situation could be a subject for future research.

**Step 1: Dimensionality reduction.** cPCA is performed on mean-centered and standard deviation-scaled data $N \times p$ matrix $X$ with the aim of finding a smaller set of linear combinations of the original $p$ variables that explains most of the variability of the data. These new variables, referred to as principal components scores, are orthogonal and uncorrelated with each other.

Let $X = (X_1', X_2', \ldots, X_N')'$ denote the full data matrix. PCA can be constructed from a singular value decomposition with $X = RDV'$, where $R$ and $V$ are orthonormal matrices and $D$ is a diagonal matrix. For an overview of PCA, see Jolliffe and Cadim [42]. The PCA scores are generated as $S \equiv RD = XV$ with the property that $S$ is an orthogonal matrix. It follows that, if $X_i \sim MVN(\boldsymbol{\mu}, \Sigma)$, then the $i$-th row of the scores matrix is also multivariate normal with $S_i =$

$X_iV \sim MVN(\mu V, V'\Sigma V)$, where $V'\Sigma V = diag\{\sigma_1^2, \ldots, \sigma_p^2\}$. Let $Y_i = S_{i,[1,k]}$ denote the first $k$ ($k \leq p$) columns of the $i^{th}$ row of $S$. Then $Y_i$ is multivariate normal with mean $\delta = (\delta_1, \ldots, \delta_k) = (\mu V)_{[1:k]}$ and variance-covariance matrix $\Omega = diag\{\sigma_1^2, \ldots, \sigma_k^2\}$.

The contribution (proportion of variance) of the $j^{th}$ PCA score column is calculated as $100\% \times \frac{d_j^2}{\sum_{j=1}^p d_j^2}$ and the *importance* of the first $k$ columns is $100\% \times \frac{\sum_{j=1}^k d_j^2}{\sum_{j=1}^p d_j^2}$, where $\{d_1, \ldots, d_p\}$ are the eigenvalues of $S$. The contribution of each subsequent column decreases such that the first principal component explains the largest variance in the data, the second principal component explains the maximum variance in the data that has not been explained by the first principal component and so on. As performed in Filzmoser et al. [31], we choose $k$ so that the importance is at least 99% of the total variance of $X$. We did explore lowering the percentage for importance, such as to 95%, but found that lower values did not result in good statistical properties (Type I error was not well-controlled). We also explored using all principal component columns ($k = p$), and observed that when $p$ is large (e.g., around 100), the Type I error was controlled, but the power dropped precipitously because of the inclusion of all the extraneous noise columns. Let $Y$ denote the reduced $N \times k$ scores matrix constructed from the first $k$ columns of $S$ and let $Y_i = (Y_{i1}, \ldots, Y_{ik})'$, denote the $i^{th}$ row of $Y$. For an inactive compound, $Y_i$ follows a multivariate normal distribution with orthogonal columns so that $Y_i \sim MVN(\delta, \Omega)$, where $\delta$ and $\Omega$ were earlier defined. The $j^{th}$ element of $Y_i$ follows a univariate normal distribution with $Y_{ij} \sim N(\delta_j, \sigma_j^2)$ and for $j \neq j'$, $Cor(Y_{ij}, Y_{ij'}) = 0$.

If both $\delta$ and $\Omega$ are known, then the Mahalanobis distance from the $i^{th}$ observation to the mean of inactives in the scores space may be calculated as

$$MD_i = \sqrt{(Y_i - \delta)' \Omega^{-1} (Y_i - \delta)} = \sqrt{\sum_{j=1}^k \frac{(Y_{ij} - \delta_j)^2}{\sigma_j^2}}.$$

**Step 2: Robust estimation of location and scale.** For the $j^{th}$ variable (i.e., $j^{th}$ column of $Y$) in the selected $k$ principal component space, a likelihood model is fitted to the data to estimate the Cauchy location parameter, which serves as a robust measure of $\delta_j$. The Cauchy distribution is symmetric at its median and has wider tails compared to the normal distribution, which allows for better accommodation of outlier-contaminated normal data. The Cauchy likelihood with location parameter $\eta$ and scale parameter $v$ for independent and identically-distributed Cauchy random variables $z_1, \ldots, z_n$ is given by

$$L(\eta, v) = \prod_{i=1}^n \frac{1}{\pi v \left(1 + \left(\frac{z_i - \eta}{v}\right)^2\right)}.$$

The estimated location parameter $\tilde{\eta}_j$ is a robust estimator for $\delta_j$. Because 68.27% of values lie within one standard deviation of the mean in a normal distribution, a robust measure for $\sigma_j$ is calculated using Motulsky and Brown's robust standard deviation of residuals [17], $rsdr_j = $ 68.27% quantile of $\{|Y_{1j} - \tilde{\eta}_j|, |Y_{2j} - \tilde{\eta}_j|, \ldots, |Y_{Nj} - \tilde{\eta}_j|\}, j = 1, \ldots, k$.

The step to robustly estimate location and scale is applied independently to each of the principal component columns. We use a Nelder-Mead [43] optimization of the log likelihood and found that the algorithm reached convergence in all of our computer simulations. For other nonlinear minimization algorithms, see [44]. Practioners of our proposed method are encouraged to check that convergence was reach for all principal component columns.

**Step 3: Outlier detection via Hotelling's T-squared testing.** Given $\tilde{\delta} = (\tilde{\eta}_1, \ldots, \tilde{\eta}_k)$ and $\tilde{\Omega} = diag\{rsdr_1^2, \ldots, rsdr_k^2\}$ as robust estimators for $\delta$ and $\Omega$, it follows under $H_0$ of (1) that

$\frac{N-k}{(N-1)k}MD_i^2 \sim F(k, N-k)$, where $MD_i^2 = (\boldsymbol{Y}_i - \tilde{\boldsymbol{\delta}})' \tilde{\boldsymbol{\Omega}}_N^{-1}(\boldsymbol{Y}_i - \tilde{\boldsymbol{\delta}})$ and $\tilde{\boldsymbol{\Omega}}_N = \tilde{\boldsymbol{\Omega}}\left(1 - \frac{1}{N}\right)$ so that $MD_i^2 = \sum_{j=1}^{k} \frac{(Y_{ij} - \tilde{\eta}_j)^2}{rsdr_j^2\left(1-\frac{1}{N}\right)}$ is a robust version of the Hotelling's $T$-squared statistic and $MD_i$ is a robust Mahalanobis distance. The testing $p$-value is

$$p-\text{value} = \Pr\left(F^* > \frac{N-k}{(N-1)k}MD_i^2\right), \qquad (3)$$

where $F^* \sim F(k, N-k)$.

As in the ROUT procedure, $p$-values from the $N$ compounds are multiplicity-corrected via the BH FDR procedure. A compound is declared active if the FDR-adjusted $p$-value is less than a pre-determined value $Q$. As in the ROUT method, we adopted $Q = 0.01$. Recall that if $\boldsymbol{Y}_i$ is not part of the calculation of the mean or variance matrix, then $\tilde{\boldsymbol{\Omega}}_N = \tilde{\boldsymbol{\Omega}}\left(1 + \frac{1}{N}\right)$ and $MD_i^2 = \sum_{j=1}^{k} \frac{(Y_{ij} - \tilde{\eta}_j)^2}{rsdr_j^2\left(1+\frac{1}{N}\right)}$. Regardless of whether $\boldsymbol{Y}_i$ is part of the calculation, with $\left(1 + \frac{1}{N}\right)$, the mROUT method is slightly more conservative and so we decided to keep $\left(1 + \frac{1}{N}\right)$ in the denominator for all calculations. For large values of $N$, the $\pm \frac{1}{N}$ term can be dropped.

## Simulations

Simulation experiments were performed to study the performance of mROUT as a function of dimension ($p$), levels of contamination (% of outliers), and distance of the outlier from the main body of data. The performance of mROUT was compared with other robust methods that have been proposed in the literature. For all simulations, the total number of observations was set to $N = 200$, which is similar to the data from our laboratories. The bulk of the observations were generated as inliers, drawn from a multivariate normal distribution. Outliers were randomly generated but located at specific Mahalanobis distances from the center of the distribution. For a given level of contamination $\varepsilon$, $N(1-\varepsilon)$ inlier observations were generated from a $p$-dimensional multivariate normal distribution with mean $\boldsymbol{\mu}$ and covariance matrix $\Sigma$, where $0 \le \varepsilon \le 0.2$, $\Sigma = diag(\boldsymbol{s}) \times \boldsymbol{C} \times diag(\boldsymbol{s})$ for correlation matrix $\boldsymbol{C}$ and vector $\boldsymbol{s}$ of marginal standard deviations. Without loss of generality, $\boldsymbol{\mu} = 0$ and all elements of $\boldsymbol{s}$ were set to 1 so that $\Sigma = \boldsymbol{C}$. The $N\varepsilon$ outlying points were generated with Mahalanobis distance $d$ from the population mean 0. See S1 Appendix for details of outlier generation procedure. Example computer code that demonstrates the simulation and mROUT procedures written in the R language [45] is provided as (S1 File).

By varying the values for $p$, $\varepsilon$, $\boldsymbol{C}$ and $d$, different simulation scenarios were created. Simulations were run with $p = 2$, 3, and 96, with $p = 96$ representing the dimension of the CRISPR assay data shown in a subsequent section. The contamination levels of outliers ranged from 0–20% of total observations. For $p = 2$, correlations were set to $\rho_{12} = 0$, 0.5, and 0.9, where $\boldsymbol{C} = \begin{pmatrix} 1 & \rho_{12} \\ . & 1 \end{pmatrix}$. For $p = 3$, three different combinations of correlations were examined with $(\rho_{12}, \rho_{13}, \rho_{23},) = (0, 0.1, 0.3)$, $(0, 0.3, 0.7)$, and $(0, 0.5, 0.7)$, where $\boldsymbol{C} = \begin{pmatrix} 1 & \rho_{12} & \rho_{13} \\ . & 1 & \rho_{23} \\ . & . & 1 \end{pmatrix}$. For $p = 96$, a single correlation matrix constructed to reflect our real data set was generated. The 96×96 correlation matrix is available as (S2 File). Except for the case when $\varepsilon = 0$, Mahalanobis distance $d$ ranged from $4 - 6$ for $p = 2$, 3, and $12 - 30$ for $p = 96$. For each scenario, 10,000 Monte Carlo were run. An overview of the values assigned to these parameters in different simulated datasets is shown in Table 1.

**Table 1. Simulation parameters.**

| Parameters | Sim 1 | Sim 2 | Sim 3 |
|---|---|---|---|
| Dimension ($p$) | 2 | 3 | 96 |
| Correlations ($C$) | $\rho_{12} = 0, 0.5,$ 0.9 | $(\rho_{12},\rho_{13},\rho_{23})$ = (0, 0.1, 0.3) (0, 0.3, 0.7) (0, 0.5, 0.7) | A $p$ x $p$ correlation matrix that mimics the structure in the real data |
| Mahalanobis distance ($d$) | 4, 4.5, 5, 5.5, 6 | 4, 4.5, 5, 5.5, 6 | 12, 13, 14, . . ., 30 |
| No. of observations ($N$) | 200 | | |
| **Contamination (100% × ε)** | 0%, 1%, 5%, 10%, 20% | | |
| No. of outliers ($N\varepsilon$) | 0, 2, 10, 20, 40 | | |

Other outlier detection procedures with available public software were evaluated and compared to mROUT. These methods come from the *rrcov* [32] and *mvoutlier* [31] libraries in R. While the work of Cabana et al. [26] looked promising, given the lack of software, we were unable to make an evaluation. From the *rrcov* library, the functions PcaCov(), PcaGrid(), PcaProj(), and PcaHubert() are rPCA-rMD methods that respectively use MCD [22], a grid-search projection pursuit [30], the projection pursuit algorithm of Croux and Riuz-Gazen [29], and the Hubert et al. algorithm that blends MCD and projection pursuit [28]. Henceforth these methods will be referred to as PcaCov, PcaGrid, PcaProj, and PcaHubert. All of the *rrcov* methods declare the $i^{\text{th}}$ observation as an outlier if the robust Mahalanobis distance $MD_i$ is greater than $\chi^2(k,0.975)$, where $k$ is the reduced dimensionality from the rPCA step. A $p$-value is easily created from this procedure as $\Pr(\chi^2(k)>MD_i)$. Based on this calculation, we tested hypotheses in (1) with BH FDR-adjusted $p$-values. From the *mvoutlier* library, the function pcout() provides the method of Filzmozer et al. [31], henceforth referred to as PCOut. Regrettably, the outlier-flagging procedure of PCOut could not be retooled into a $p$-value or any other probability metric; thus, we could not adjust it to avoid a high false positive rate.

For each simulated scenario and outlier detection procedure, Type I error, false discovery rate, and power were estimated. Type I error was estimated for inlier observations with $\varepsilon = 0$ by the proportion of times (out of 10,000) the unadjusted $p$-value was less than or equal to $Q = 0.01$. For mROUT, the unadjusted $p$-value is given by Eq (3) and for *rrcov* procedures, the unadjusted $p$-value is given by $\Pr(\chi^2(k)>MD_i)$. For PCOut, because there are no $p$-values, the proportion of outlier declarations was used to estimate Type I error. For the evaluation of hypotheses in (1), $H_1$ was declared (the observation is an outlier) whenever the BH FDR-corrected $p$-value was less than or equal to $Q = 0.01$. For PCOut, $H_1$ was declared whenever the software flagged an observation as an outlier. A false positive is declared whenever an inlier is misclassified as an outlier. A true positive is declared whenever an outlier is correctly identified. For scenarios with $\varepsilon > 0$ and $d > 0$, the false discovery rate (FDR) = FP/(FP+TP) was calculated for each Monte Carlo run, where FP = number of false positives and TP = number of true positives out of $N = 200$ observations. The reported FDR is the Monte Carlo average over 10,000 simulated runs. Adjusting for Monte Carlo error, both Type I error and FDR can be as high as 0.013. Scenarios with Type I error and FDR less than 0.013 were considered to be in control. Finally, statistical power of mROUT and *rrcov* procedures was calculated as the proportion of times (out of 10,000) the BH FDR-adjusted $p$-value was less than or equal to $Q = 0.01$ for a single outlier observation in scenarios with $\varepsilon > 0$ and $d > 0$. For PCOut, unadjusted statistical power was calculated based on the proportion of contaminated observations correctly flagged as outliers.

## Results

### Low-dimensional simulation results

Although mROUT was developed primarily for hit screening in high-dimensional data, we also evaluated its performance against the outlier detection algorithms mentioned in previous sections in 2- and 3-dimensional settings to observe relationship between the simulated outcomes and correlation.

The behaviours of mROUT, PcaCov, PcaHubert, PcaGrid, PcaProj and PCOut were examined in the absence of outliers for $p$ = 2, 3. As shown in Fig 2A and 2B, mROUT, PcaCov and PcaHubert exhibited Type I errors that are within the acceptable range in both dimensions, independent of the correlation structure of the data. For the two projection pursuit-based approaches, PcaProj and PcaGrid, Type I error appear to be under control in most cases but starts to elevate as correlation between variables increases. PCOut, however, exhibited uncontrolled Type I error of roughly 0.12 across all simulation settings, more than 10x the nominal threshold $Q$ = 0.01 (see S1 Table).

Fig 2C and 2D display the FDR of mROUT against the *rrcov* procedures for $p$ = 2, 3 and $Q$ = 0.01 at different contamination levels. FDR is well-controlled for mROUT, reasonably controlled for PcaCov and PcaHubert (very slightly elevated in some cases), and sometimes out of control for the projection-pursuit methods PcaGrid and PcaProj. The FDR of *rrcov* procedures appear to be influenced by the amount of contamination and correlation structure, where the worst behaviour is observed for PcaGrid and PcaProj in scenarios with 1% contamination that involves variables with larger correlation. Because PCOut could not be adjusted for multiple testing, the FDR for PCOut is a decreasing function of the contamination level ($\varepsilon$) and ranged between 0.12–0.92 in our simulations (see S2 Table).

Lastly, the statistical power to detect the effects of contamination level $\varepsilon$ and Mahalanobis distance $d$ was examined. The results are presented in Fig 2E and 2F, which display power as a function of $\varepsilon$ in different correlation settings with $d$ ranged from $4 - 6$, $p$ = 2, 3 and $Q$ = 0.01. Two features are apparent from these plots. First, for a specific contamination level, the Mahalanobis distance required for confidently declaring an outlier increases as the dimension rises. Second, as the amount of contamination rises beyond 10%, all methods suffer a drop in power and only outliers with $d \geq 5.5$ were almost always detected. Out of all the methods, the best overall performance is observed for mROUT as it attains reasonably high power (99% for $p$ = 2, 94% for $p$ = 3) in identifying outliers with $d \geq 5.5$ without being affected by the correlation structure of the data. When the level of contamination rises to 20%, PcaCov and PcaHubert seem to outperform mROUT in cases with smaller $d$, but their respective statistical power drops markedly in highly correlated data. Similarly, PcaGrid and PcaProj show comparable power to mROUT in most cases but perform less well in scenarios with highly correlated variables; however, the increase in power for PcaGrid and PcaProj may be partially attributed to their inflated Type I error and FDR. PCOut has exceptionally high power (~100%) in all scenarios; however, this comes at the cost of uncontrollable FDR, meaning that the user cannot be sure if a detected outlier is a true or false positive (S2 and S3 Tables).

### High-dimensional simulation results

Based on the results obtained from the low-dimensional simulation runs, we selected three methods that showed a good balance in performance and computational speed for evaluation in higher dimensional setting. The chosen approaches are mROUT, PcaHubert and PcaProj. While the MCD-based PcaCov approach showed good overall performance in lower

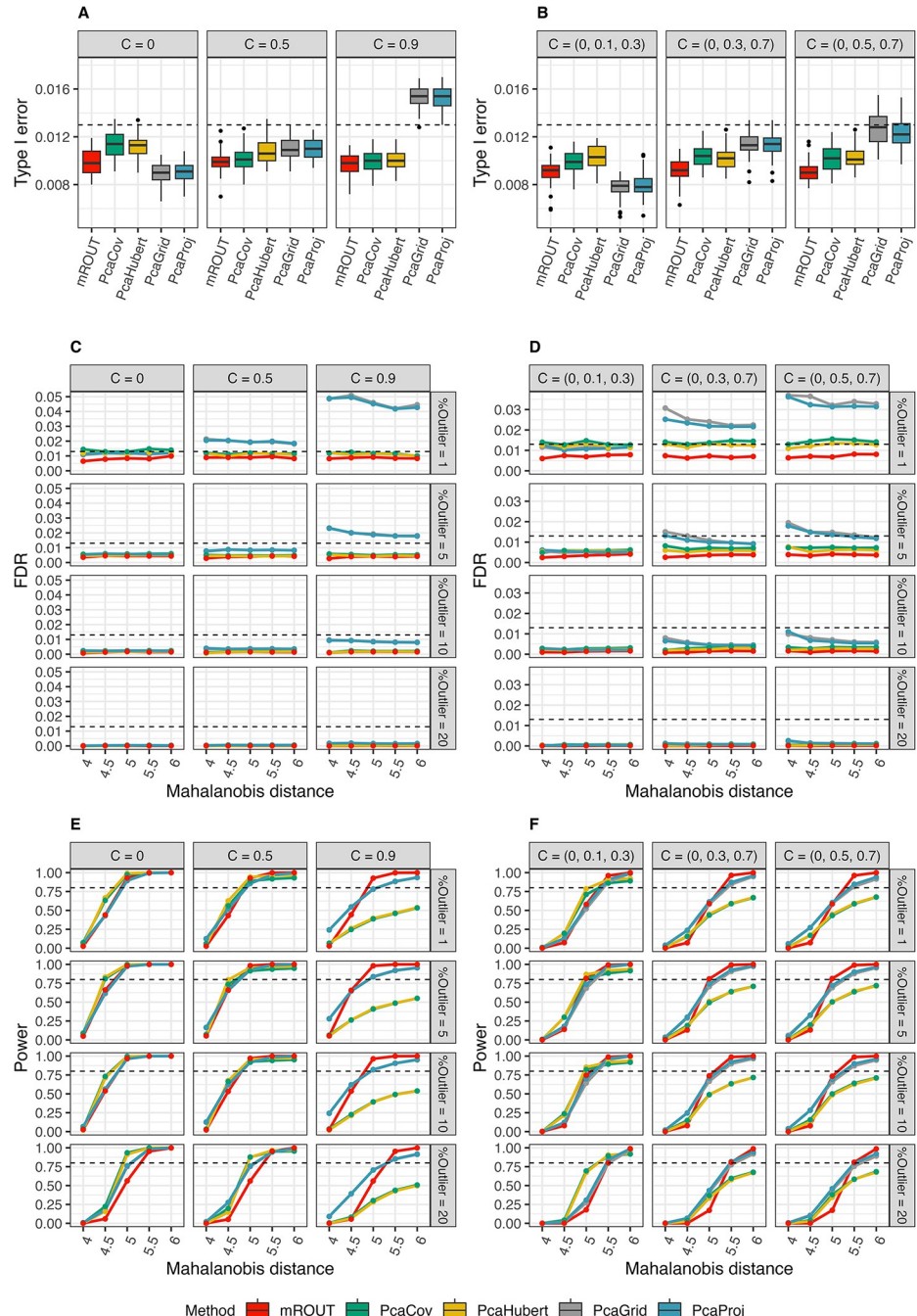

**Fig 2. Performance of mROUT and selected rPCA outlier detection procedures on 2- and 3-dimensional simulated data.** (A) Type I error for $p = 2$, $\varepsilon = 0$, (B) Type I error for $p = 3$, $\varepsilon = 0$, (C) FDR for $p = 2$, (D) FDR for $p = 3$, (E) power for $p = 2$, and (F) power for $p = 3$. The performance measures were estimated through 10,000 simulations with $N = 200$. Dashed line in (A), (B), (C) and (D) represents Monte Carlo error at 0.013. Dashed line in (E) and (F) represents 80% power.

dimensional space, its runtime (for us) was prohibitively high for $p = 96$. We did not apply PcaGrid because its performance was similar, but slightly inferior to PcaProj. Finally, we did not include PCOut given its Type I error and FDR performance.

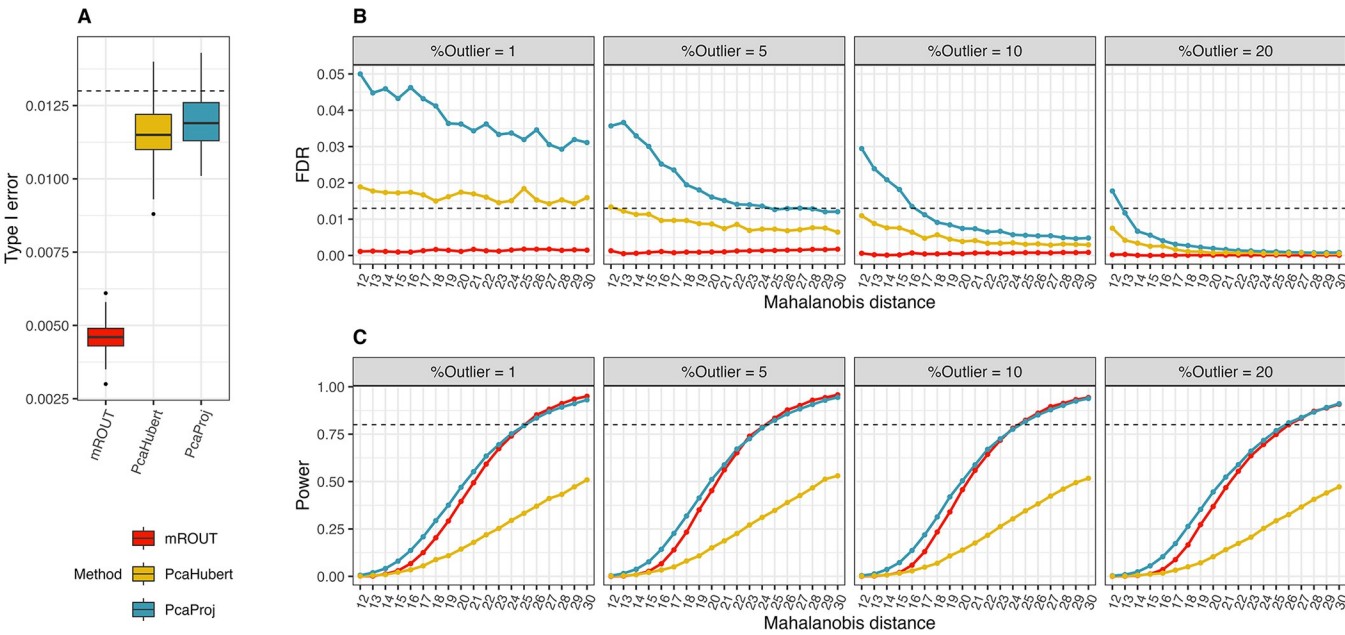

**Fig 3. Performance comparison of mROUT, PcaHubert and PcaProj on 96-dimensional simulated data.** (A) Type I error, (B) FDR, and (C) power were estimated through 10,000 simulations for each combination of ε and $d$ with $N = 200$. Dashed line in (A) and (B) represents Monte Carlo error at 0.013. Dashed line in (C) represents 80% power.

The simulation was performed with $p = 96$, $N = 200$ and a single correlation matrix that mimics the structure of a real data set. The levels of contamination are identical to those explored in previous simulation experiments, but the range of Mahalanobis distance $d$ (in 96-dimensional space) was increased to $12 - 30$.

All three methods have reasonably good performance under the null hypothesis of no outliers in (1) and maintain the Type I errors at an acceptable level ($< 0.013$). Notably, mROUT has the lowest Type I error while PcaHubert and PcaProj are slightly elevated (Fig 3A). There is strong suggestion that PcaProj generates an overage of false positives in systems with low level of contamination ε = 1% and 5%. PcaHubert displays slightly elevated FDR in scenarios with ε = 1% (Fig 3B). mROUT is the only approach that keeps the FDR under control below $Q = 0.01$ across all settings, an observation that is consistent with the results from $p = 2, 3$ simulation studies. In terms of statistical power, mROUT and PcaProj show comparable performance and both outperform PcaHubert for the range of ε and $d$ tested, but the FDR rate of PcaProj can sometimes be undesirably large (Fig 3B and 3C and Table 2). For a given contamination level, mROUT and PcaProj were able to detect outliers that are at least 26 Mahalanobis units away from the center of the data with 80% power, and the power remained relatively constant as the level of contamination grows from 1% to 20%.

## Intrinsic performance evaluation

In the comparative analysis presented in the previous section, the number of observation ($N$) was fixed at 200 to emulate the size of a typical data set encountered in hit screening assays from our laboratories. Using the same simulation setup, we extended our experiments to assess the outlier detection efficacy of mROUT in system with fewer observations, where $N$ was dropped to 20, 40 and 100. We found that, while Type I error and FDR are maintained at levels below 0.01, there is a noticeable decrease in statistical power when the number of observations

**Table 2. Statistical power and false discovery rate (FDR) of mROUT, PcaHubert and PcaProj in high-dimensional simulations using Q = 0.01.**

| p | ε (%) | d | Power | | | FDR | | |
|---|---|---|---|---|---|---|---|---|
| | | | mROUT | PcaHubert | PcaProj | mROUT | PcaHubert | PcaProj |
| 96 | 1 | 26 | 0.852 | 0.370 | 0.835 | 0.002 | *0.015* | *0.035* |
| | | 28 | 0.911 | 0.432 | 0.893 | 0.001 | *0.015* | *0.029* |
| | | 30 | 0.950 | 0.509 | 0.931 | 0.001 | *0.016* | *0.031* |
| | 5 | 26 | 0.878 | 0.390 | 0.857 | 0.001 | 0.007 | *0.013* |
| | | 28 | 0.929 | 0.467 | 0.907 | 0.002 | 0.008 | *0.013* |
| | | 30 | 0.958 | 0.530 | 0.944 | 0.002 | 0.006 | *0.012* |
| | 10 | 26 | 0.861 | 0.382 | 0.851 | 0.001 | 0.003 | 0.005 |
| | | 28 | 0.912 | 0.460 | 0.902 | 0.001 | 0.003 | 0.005 |
| | | 30 | 0.943 | 0.517 | 0.938 | 0.001 | 0.003 | 0.005 |
| | 20 | 26 | 0.800 | 0.326 | 0.810 | 0.000 | 0.001 | 0.001 |
| | | 28 | 0.871 | 0.406 | 0.866 | 0.000 | 0.001 | 0.001 |
| | | 30 | 0.908 | 0.472 | 0.910 | 0.000 | 0.001 | 0.001 |

Cases with FDR greater the nominal threshold $Q = 0.01$ are denoted in italics.

Taken together, mROUT achieves the best overall performance in both low- and high-dimensional simulation settings. It outperforms other rPCA-rMD-based outlier detection procedures in terms of controlling Type I error and FDR while attaining good power in system with up to 20% of contaminations.

is limited ($N = 20$ and 40), even when $p = 2$ and 3. The decline in performance is likely due to insufficient data for accurately estimating the inactive distribution and robust statistics. As the number of observations increases ($N = 100$), there is a marked improvement in statistical power, irrespective of the number of dimensions and the level of contamination in the system (see S3 File).

Controlling the FDR is an important consideration for multivariate outlier detection because, in practice, the candidate outliers are not known in advance, and all observations are tested sequentially. Up to this point, we have adhered to the $Q$ value recommended by Motulsky and Brown in the original ROUT paper [17]. By setting $Q = 0.01$, we are aiming to restrict the 'hits' to observations that fall beyond the 99[th] percentile of the inactive distribution. In certain applications, a different $Q$ value may be preferable.

To provide insights into the trade-offs between true positive rate (sensitivity) and false positive rate (1-specificity) of our proposed method, we used simulated data to evaluate the effects of varying $Q$ within the range of 0.001 to 1. The simulations were conducted with $p = 96$, $N = 200$, incorporating different levels of contamination at 1%, 5%, 10% and 20%. Mahalanobis distance (MD) values of 15, 20, 25 and 30 with reasonable discovery rate were used. In each scenario, 10,000 Monte Carlo runs were performed to estimate the true positive rate (TPR) and false positive rate (FPR). The simulation outcomes are presented as receiver operating characteristic (ROC) curves in Fig 4. The results show that mROUT effectively constrains the false positive rate (FPR) across the range of tested $Q$ values. While TPR (sensitivity) remains low when MD = 15 (represents small effect size that is difficult to detect), it rises to over 75% when MD exceeds 25, without compromising the FPR. In all the scenarios tested, increasing $Q$ from 0.01 to 0.05 lead to an improvement in the TPR, but has minimal impact on the FPR. These findings suggest that mROUT is well-suited for applications where prioritizing higher sensitivity is advantageous, as it alleviates concerns about incurring an excessively high false positive rate when using a higher $Q$ value.

In the next section, we will demonstrate the performance of mROUT on a real data set generated from an in-house phenotypic screen.

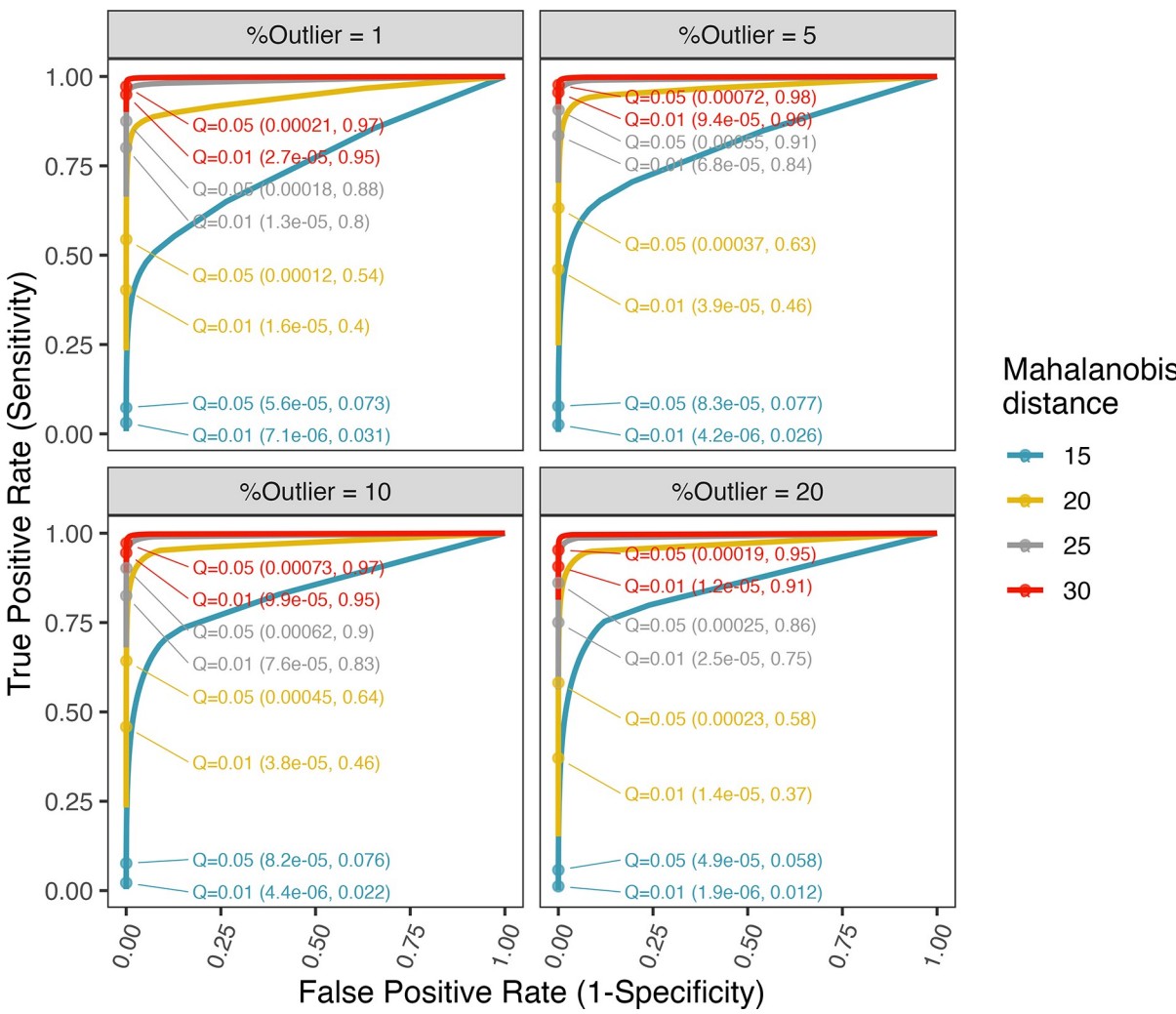

**Fig 4. ROC curves of mROUT obtained by varying the significance threshold Q.** False positive rate (x-axis) is calculated as FP / (FP + TN), where FP is the number of false positives and TN is the number of true negatives. True positive rate (y-axis) is calculated as TP / (TP + FN), where TP is the number of true positives and FN is the number of false negatives. The numbers in the brackets indicates the false positive rate (first number) and true positive rate (second number) observed for $Q$ = 0.01 or 0.05.

## Application to real data

Although hit screening was described as a method to detect the activity of compounds, it can be viewed more generically, such as in this high-content image-based CRISPR screening data set (provided as S4 File). This data set was collected as part of a high-throughput arrayed CRISPR knockout screening effort for identifying potential novel drug targets for chronic kidney disease (CKD) in a disease-relevant renal cell model–primary human glomerular microvascular endothelial cells (HGMECs, Cell Systems, catalog no. ACBRI 128). A gene knockout that produced the desired phenotypic changes in HGMECs upon treatment with pro-inflammatory cytokines is considered a "hit". In this study, the activity of single-shot gene knockouts was examined relative to inactive and active controls. The entire experiment was supported on a single 384-well plate with 218 gene knockouts (KO), 14 inactive control (IC) wells, 14 active control (AC) wells, and 6 lethal control (LC) wells. A priori, it was assumed that the majority of knockouts would not show activity.

Images of the wells were acquired using a Cell Voyager 7000S spinning disk confocal microscope (Yokogawa) and processed with a multi-task convolutional neural network (CNN). The CNN was trained on a self-supervised task using contrastive learning to represent the image content [46] with a binary classification task to differentiate between inactive and active controls. The two tasks are trained jointly by adding the loss functions from each task to give a univariate loss to be minimized. We use the output from the classification task to assign cells in each well as active or inactive, and then take the fraction of active cells in each well as the standard univariate response variable. The data were initially analyzed with a univariate statistical approach but we found that it lacks the sensitivity to detect gene knockouts showing subtle and distinct phenotypic changes. Recognizing that CKD is a complex disease with multiple patient endotypes, a more sophisticated approach that scores the phenotype beyond simple classification was considered. Instead of analyzing the results from the final classification node of the neural net, we captured $p = 96$ dimensional data from the penultimate node and took the median across cells in each well to give a single vector per treatment well. In this high dimension, the data have no intrinsic meaning, but given the self-supervised neural net training, should separate gene knockouts that show an activity different to the inactive controls based on their visual appearance in the images.

The mROUT procedure was applied to the 218 KOs and 14 ICs from the $p = 96$ data. Before PCA, the data were mean-centered and standard deviation-scaled. The number of PCA components required to explain 99% of variability in the original data was $k = 11$. Using a significance threshold $Q = 0.01$, mROUT identified 12 gene KOs as hits. All the ACs and LCs were detected (true positives) and none of the ICs were declared as hits (true negatives). Fig 5A shows a two-dimensional PCA plot of the study data with the PCA model used to project the

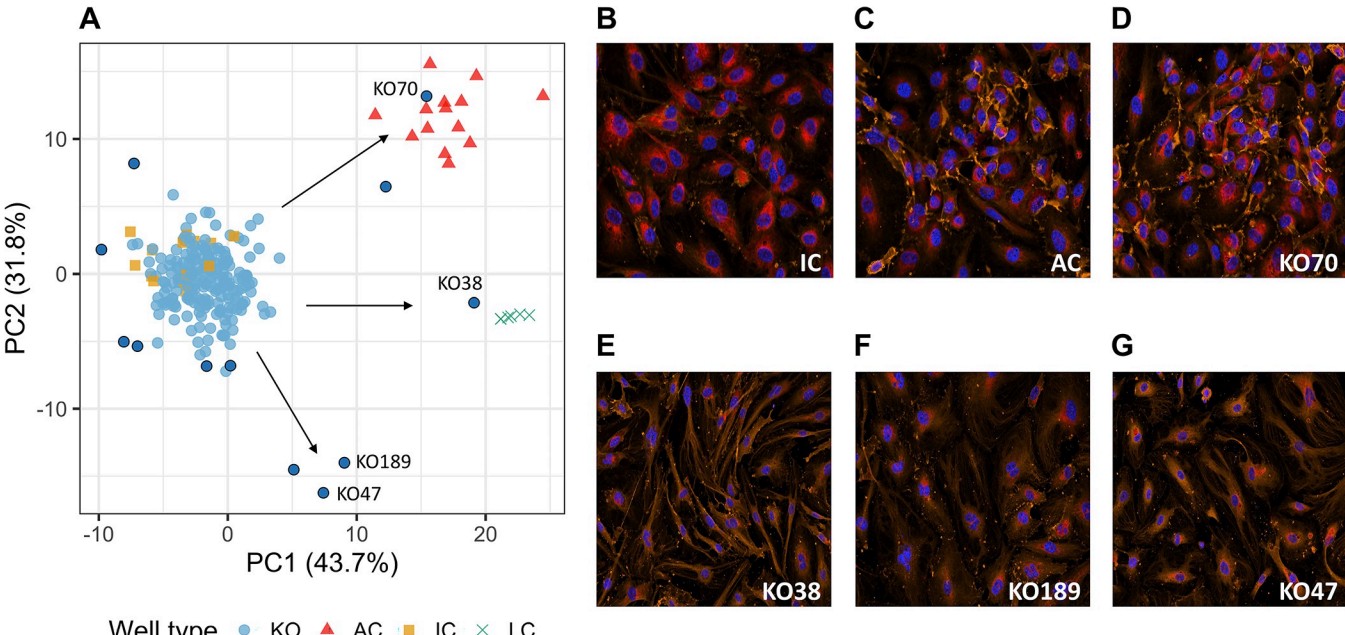

**Fig 5. mROUT was able to identify biologically relevant hits in real data set.** (A) Two-dimensional representation of the feature space data. Acronyms are KO = gene knockout, AC = active control, IC = inactive control, LC = lethal control. Dark blue circles represent observations detected by mROUT as potentially active KOs. (B) to (G) are confocal images of phenotypes seen in IC, AC and four active KOs detected by mROUT. IC showed stressed phenotype induced by cytokine treatment. AC showed healthy phenotype. KO70 overlaps with the AC in feature space and showed similar phenotype as the AC. KO38, KO189 and KO47, which deviates from the IC-AC axis, showed fibrotic- and apoptotic-like phenotypes suggestive of novel disease-driving mechanisms. The cells were stained with CD31 (orange), Hoechst (blue) and CellMask (red).

AC and LC wells. Most of the gene KOs (light blue circles) were mapped to the same cluster as the IC (orange squares) with the detected KOs (dark blue circles) appearing to be detached from the main cluster and pointed towards different directions.

While it is beyond the scope of this paper to delve into the specific biology associated with each of the detected KOs, the two active KOs pointed in direction of the ACs are likely the most biologically relevant hits because they were able to prevent the cytokine-induced stressed phenotype seen in the ICs while overlapping with the feature space of ACs (Fig 5B–5D). It may, however, also be worth exploring other active KOs that deviated from the IC-AC axis, because some of them showed distinct phenotypes that could be associated with novel molecular mechanisms driving the disease (Fig 5E–5G).

Instead of relying solely on a two-dimensional visualization, one can enrich the mROUT results by additionally calculating a multivariate angle for each KO relative to the means of IC and AC. Angle calculations will be briefly explored in the discussion section.

## Discussion

The mROUT procedure was illustrated through simulation and a real data set, and shown to detect outliers with statistical power competitive with rPCA-rMD methods while also maintaining Type I error and FDR. Thus, mROUT can be a powerful tool for multidimensional hit screening. There are several considerations for future work, which are discussed below.

The neural network architecture we have implemented tends to generate approximately normally distributed features in our hands, however for other morphological profiling features, such as those generated from CellProfiler [47], this is usually not the case. To enable application of mROUT to such data, we propose incorporating a variational autoencoder (VAE) [48,49] into the image feature extraction workflow. The VAE serves to enforce the distribution of the latent space alongside the reconstruction loss through the inclusion of the Kullback-Leibler divergence term in the overall loss function. This regularization encourages the VAE to learn a latent space adhering to a specific distribution—in our case, a standard normal distribution—while simultaneously optimizing for precise data reconstruction.

As implied in the real data section, the relative angle between a gene KO and two reference points (e.g., inactive and active controls) may provide additional information to enrich or perhaps triage the list of detected KO activity. In terms of mathematical vector properties, the mROUT system can detect outliers that are of a certain magnitude with no regard for direction. If a system contains two reference points, such as the robust mean of the inactives (from the Cauchy step of mROUT) and mean of active controls in the PCA scores space, the angle of a KO can be relatively calculated. Let $\boldsymbol{\mu_0}$ and $\boldsymbol{\mu_1}$ denote the robust inactives mean and active control mean in the scores space, and let $\boldsymbol{y}$ denote the scores vector for a KO of interest. The angle of interest between $\boldsymbol{y}-\boldsymbol{\mu_0}$ and $\boldsymbol{\mu_1}-\boldsymbol{\mu_0}$, where the angle in radians between two vectors $\boldsymbol{u}$ and $\boldsymbol{v}$ is given by $\operatorname{acos}\left\{\dfrac{(\sum_{j=1}^{k} u_j v_j)}{\sqrt{\sum_{j=1}^{k} u_j^2 \times \sum_{j=1}^{k} v_j^2}}\right\}$. By this calculation, the direction may also be considered in hit screening.

The quality of an assay run should be examined prior to implementing mROUT for hit detection. Given a system with a set of inactive controls and active controls, one can calculate a multidimensional analogue to the Strictly Standardized Mean Difference (SSMD) quality statistic proposed by Zhang [50] for low and high univariate controls, given by $SSMD = \dfrac{\mu_H - \mu_L}{\sqrt{\sigma_L^2 + \sigma_H^2}}$, where $\mu_L$ and $\mu_H$ denote low and high control means and $\sigma_L$ and $\sigma_H$ denote the low and high control standard deviations. A Mahalanobis distance version of SSMD for multivariate data is

given by

$$MD_{qc} = \sqrt{(\boldsymbol{\mu}_{C2} - \boldsymbol{\mu}_{C1})'(\boldsymbol{\Sigma}_{C1} + \boldsymbol{\Sigma}_{C2})^{-1}(\boldsymbol{\mu}_{C2} - \boldsymbol{\mu}_{C1})}$$

where $\boldsymbol{\mu}_{C1}$ and $\boldsymbol{\mu}_{C2}$ denote the means and $\Sigma_{C1}$ and $\Sigma_{C2}$ denote the variance-covariance matrices of the two controls. Quality control cut-offs for $MD_{qc}$ can be determined empirically through observations from assay development and its remaining assay life cycle.

Finally, we consider the issue of combining data from multiple plates. Possible solutions include normalization to remove the plate effect and modelling the plate effect in one of the mROUT steps. In our process, the image data is normalized to the [0, 1] range prior to the neural net step to remove gross plate-to-plate differences in image intensity. We also apply a plate mean subtraction step at the $p$-dimensional level (i.e., the raw data). If additional location effects are suspected, one may also correct for spatial effects (i.e., systematic row and column effects on the plate) using normalization techniques, such as B-score [11] or Loess [51] corrections. Alternatively, a plate, row, and column terms could be included in the PCA or the Cauchy step of the mROUT procedure. We have not fully explored the pros and cons of each plate adjustment step and view these corrections as worthy of further consideration.

## Conclusions

We propose a new method mROUT for identifying outliers in multivariate assays based on principal components analysis and robust Mahalanobis distance. Analyses of computer-simulated data demonstrate that mROUT attains high true discovery rates while controlling Type I error and FDR. Relative to competitor procedures found in the literature with publicly available software, mROUT is the only method that include an FDR controlling step, implying that, without our modifications, the competitor methods do not control FDR. Even with our modifications, the competing procedures sometimes produce FDR that are larger than expected. When applied to real phenotypic screening data, mROUT was able to identify biologically relevant hits and uncovered potential novel mechanistic insights through genotype-phenotype mapping. Thus, for multivariate assays with multivariate normal data, such as the neural net-processed CRISPR image data in our example, mROUT can be a powerful tool for hit screening. Future work may involve comparisons to techniques incorporating a nonlinear dimension reduction step, while also exploring their applicability to non-Gaussian data.

## Supporting information

**S1 Appendix. Outlier generation procedure.** Details of the outlier generation method used in the simulation study.
(DOCX)

**S1 File. Computer code.** Computer code written in the R programming language that illustrates the mROUT procedures on a simulated multivariate data set.
(R)

**S2 File. Correlation matrix used in high-dimensional simulations.** The 96×96 correlation matrix used in the simulation study with $p$ = 96.
(CSV)

**S3 File. Effect of sample size.** Performance of mROUT estimated for 2-, 3- and 96-dimensional simulations with different number of observations.
(DOCX)

**S4 File. CRISPR screen data.** Real CRISPR screening data that contains $p = 96$ features from the penultimate node of the neural net.
(CSV)

**S1 Table. Type I error of simulation studies.** Type I error of mROUT and other outlier detection methods estimated for 2-, 3- and 96-dimensional simulations.
(DOCX)

**S2 Table. FDR of low-dimensional simulation studies.** FDR of mROUT and other outlier detection methods estimated for 2- and 3-dimensional simulations.
(DOCX)

**S3 Table. Power of low-dimensional simulation studies.** Statistical power of mROUT and other outlier detection methods estimated for 2- and 3-dimensional simulations.
(DOCX)

## Author Contributions

**Conceptualization:** Steven Novick.

**Formal analysis:** Hui Sun Leong, Adam Corrigan, Steven Novick.

**Methodology:** Hui Sun Leong, Tianhui Zhang, Adam Corrigan, Steven Novick.

**Resources:** Alessia Serrano, Ulrike Künzel, Niamh Mullooly, Ceri Wiggins.

**Software:** Hui Sun Leong, Tianhui Zhang, Adam Corrigan, Steven Novick.

**Validation:** Hui Sun Leong, Steven Novick.

**Visualization:** Hui Sun Leong, Steven Novick.

**Writing – original draft:** Hui Sun Leong, Steven Novick.

**Writing – review & editing:** Hui Sun Leong, Tianhui Zhang, Adam Corrigan, Yinhai Wang, Steven Novick.

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
