## [Decision Letter · Decision Letter 0]

13 Jun 2024

PONE-D-24-10015Hit screening with multivariate robust outlier detectionPLOS ONE

Dear Dr. Leong,

Thank you for submitting your manuscript to PLOS ONE. After careful consideration, we feel that it has merit but does not fully meet PLOS ONE’s publication criteria as it currently stands. Therefore, we invite you to submit a revised version of the manuscript that addresses the points raised during the review process.

We look forward to receiving your revised manuscript.

Kind regards,

Longxiu Huang, Ph.D.

Academic Editor

PLOS ONE

Journal Requirements:

Reviewers' comments:

Reviewer's Responses to Questions

**Comments to the Author**

1. Is the manuscript technically sound, and do the data support the conclusions?

Reviewer #1: Yes

Reviewer #2: Yes

Reviewer #3: Partly

2. Has the statistical analysis been performed appropriately and rigorously? 

Reviewer #1: N/A

Reviewer #2: Yes

Reviewer #3: No

3. Have the authors made all data underlying the findings in their manuscript fully available?

Reviewer #1: Yes

Reviewer #2: Yes

Reviewer #3: Yes

4. Is the manuscript presented in an intelligible fashion and written in standard English?

Reviewer #1: Yes

Reviewer #2: Yes

Reviewer #3: Yes

5. Review Comments to the Author

**Reviewer #1:** See the attached file for the full comments. One major comment is there. In the dimensional reduction step, it's mentioned that at least $99\\%$ of the total variance of $X$ are retained. I'm interested in knowing the value of $k$ in the experiments, particularly for the high-dimensional case (p=96). I'm curious whether the orthogonality of $Y$ is necessary while the dimensional reduction might be unnecessary for mROUT. Could the authors test mROUT with full dimensions? If such comparisons have already been made, it would be beneficial to discuss the necessity of dimensional reduction in mROUT, although the discussion of its necessity in other methods is already included.

**Reviewer #2: **This paper describes a multivariate outlier detection method (mROUT) that combines PCA with robust Mahalanobis distance computation. Some of the key steps include estimating the Cauchy-Lorentz location parameter using maximum likelihood (Levenberg–Marquardt DLS method) and estimating the scatter using the Motulsky-Brown robust standard deviation of residuals (rsdr). The authors have demonstrated in their results the importance of incorporating BH multiplicity-adjustment on the p-value to control the false detection rate (FDR) when outlier detection is conducted via T-squared testing. The motivation and essential details of the proposed algorithm are clearly described. Although the results demonstrate quite convincingly the strengths of the proposal within the current scope of the experiments, further discussion to elucidate the merits and general limitations of the approach should be encouraged. Perhaps the authors can consider re-parameterising/extending some of the experiments and including relevant findings where appropriate. More detailed suggestions are given in particular in Comments 4-8.

Overall, this work represents an original research contribution to the field of phenotypic screening / biostatistics. Specifically, the authors have developed an FDR-calibrated multivariate outlier detection approach that may facilitate more effective identification of potential targeted treatments. In practical terms, it may lead to discovery of compounds for the treatment of complex diseases, and gaining insights into novel pathways or mechanisms. The reviewer is intrigued to see some of the phenotypic changes following treatment and the use of convolutional neural network, self-supervised contrastive learning in a binary classification task to represent well images and differentiate between inactive and active controls; then applying outlier detection to features extracted from the penultimate layer of a CNN. The results (e.g. line 399-406) show tremendous promise. The reviewer hopes this can be scaled up and adopted more widely.

List of Comments

[Comment 1] Introduction (L39-40)

1a) Regarding normally distributed inactives [as observed in multi-dimensional cell painting or morphological profiling] - how often is this a valid assumption?

1b) What methods are available for handling situations where the Gaussian assumption is violated? PCA seems like a reasonable choice when there is no preferential direction (all variables are valued equally) or non-linearity observed in the point distribution. When this is not true, have the authors considered using local neighbourhood embedding [1] or kernel techniques [2] to handle non-Gaussian data for the inactives? Any thoughts regarding the suitability of manifold learning methods?

1c) A statistical comparison of outlier detection efficacy - with and without these embedding, manifold learning or kernel techniques - may be considered as part of future work.

[1] Jiaqi Xue, Bih Zhang, Qianyao Qiang. Local Linear Embedding with Adaptive Neighbors, Pattern Recognition, vol. 136, article 109205, Apr 2023. DOI: https://doi.org/10.1016/j.patcog.2022.109205

[2] P. Rudra, R. Baster, E. Hsieh, D. Ghosh. Compositional data analysis using kernels in mass cytometry data, Bioinformatics Advances, vol. 2., issue 1., 2022. DOI: https://doi.org/10.1093/bioadv/vbac003

[Comment 2] Multivariate outlier detection (L158-161)

The authors observed that "even after we modified rPCA-rMD methods to also provide a BH FDR-adjusted p-value..., we found that our method had better Type I error control, better FDR control, and/or higher statistical power to declare H1 in (1) compared with other methods." Can you offer some explanation as to why these other techniques produced inferior results despite the p-value FDR adjustment?

[Comment 3] mROUT algorithm, step 2

3a) (L208-209) Was the Levenberg–Marquardt nonlinear regression algorithm used to estimate the Cauchy-Lorentz location parameter? If so, please mention and include a reference (e.g. [4]). What are its competitive advantages?

[4] H.P. Gavin, The Levenberg-Marquardt algorithm for nonlinear least squares curve-fitting problems, 2024. DOI: https://people.duke.edu/~hpgavin/ExperimentalSystems/lm.pdf

3b) Can you provide some information about the optimizer used to maximise the likelihood. In terms of practicality, any particular reason for using the Nelder and Mead (1965) method, or not using quasi-Newton and L-BFGS-B.

3c) Did you encounter any difficulty estimating the Cauchy parameters using real data.

3d) Have you implemented any checks to ensure the estimates are sensible, i.e., detect anomalies or reject poor estimates. Do you envisage any situation where the maximum likelihood technique might fail *silently* when deployed in a large scale setting?

[Comment 4] Simulation (L237)

An important question is whether the performance of the proposed method would hold for fewer observations (e.g. N between 20 and 50)?

Recommendation 1 - Where possible, the experiments should be repeated for different N, and graphs plotted to observe any changes in the results.

[Comment 5] Application to real data (L402-403) "Using a significance threshold Q=0.01, mROUT identified 12 gene KOs as hits. All ACs and LCs were detected and none of the ICs were declared as hits"

5a) This is encouraging. It would be helpful to provide some guidance on the selection of Q, if we want to view this as a parameter tuning exercise.

Recommendation 2 - Visualise in a graph how statistical power (sensitivity) and FDR vary with Q. This will help readers outside a clinical setting to appreciate the type 1 and type 2 errors trade-off (often the F1 score is skewed depending on the application, sometimes higher recall is preferred over precision).

General Questions - not tied to a specific section

[Comment 6] In chemometrics, data can have dimensions in the thousands. Is there evidence that the proposed technique would work equally well for this case (where the number of variables p >> 96)? Would it make sense to use projection pursuit initially to retain the most critical information (by whatever criteria) and reduce the number of dimensions to something more manageable?

The concern is that PCA explains data variance using eigenvectors aligned with directions of maximum variation. This is optimal for linear decomposition. However, the reliability of this procedure depends on the proportion and magnitude of the outliers (relative to the inactives). Second, the Euclidean metric is only meaningful if important differences are reflected by large distances as measured by the L2 norm. Are the authors aware of any situation where nonlinear dimensionality reduction would be of benefit, where manifold learning would be needed to preserve small differences or latent structures of biological significance?

[Comment 7] What are some of the known limitations of the proposed method? For example, for tiny data sets, how would it behave if applied in another problem domain where the number of measurements N is small (e.g. N between 20-50, rather than > 200)? This is related to Comment 4.

Can this be explored through Monte Carlo simulation, by randomly drawing say {25, 50, 100} samples from the complete set of measurements (which acts as ground truth) to emulate such effect? The reviewer is thinking how this would affect the multiplicity-adjusted p values and hypothesis testing outcomes.

[Comment 8] How would the proposed method behave given perfect data (with no outliers) in high dimensions (let's say the retained PCA components k far exceeds 11), and garbage data of a psuedo-random nature with no trends at all. Does the current design raise a flag, to perhaps alert scientists that the data is unexpected or perhaps some error has been introduced during the experiments?

**Reviewer #3: **A new multivariate robust outlier identification technique is presented in this paper. Despite the appealing idea and promising methodology, the discussions, conclusions, and result interpretation have some significant shortcomings. Additionally, there are major improvements in writing and presentation required. Here are some main significant problems (from my point of view)

1- The literature review and proper citations are missing. It is necessary to update the References. The references are extremely old, with several dating back to before 2010, and there is just one reference from 2021.

2- The existing techniques for detecting outliers multivariate data are not well reviewed. Numerous recent papers have presented robust techniques for the identification of outliers. A comparative analysis of the offered new approach in this study and existing techniques in recent papers is significantly needed.

Here are examples of few recent papers:

- Vishwakarma et al., 2021. A hybrid feedforward neural network algorithm for detecting outliers in non-stationary multivariate time series. Expert Systems with Applications 184:115545

- Touny et al., 2024. Scalable fuzzy multivariate outliers identification towards big data applications. Applied Soft Computing 155: 111444

- Vishwakarma et al., 2023. An automated robust algorithm for clustering multivariate data. Journal of Computational and Applied Mathematics 429: 115219

- Hilal et al., 2022. FinanciaL Fraud: A Review Of Anomaly Detection Techniques And Recent Advances. Expert Systems with Applications

193: 116429

3- The main article must include tables that present some performance measures to compare the new method and the existing techniques.

4- The organization and presentation of the results, along with a lack of clear interpretation are major challenges. There should be some discussion on the theoretical arguments for the new method's superiority.

5- There is a need to enhance the resolution of the figures.

6. PLOS authors have the option to publish the peer review history of their article (what does this mean?). If published, this will include your full peer review and any attached files.

Reviewer #1: No

Reviewer #2: No

Reviewer #3: No

---

## [Decision Letter · Decision Letter 1]

12 Aug 2024

PONE-D-24-10015R1Hit screening with multivariate robust outlier detectionPLOS ONE

Dear Dr. Leong,

Thank you for submitting your manuscript to PLOS ONE. After careful consideration, we feel that it has merit but does not fully meet PLOS ONE’s publication criteria as it currently stands. Therefore, we invite you to submit a revised version of the manuscript that addresses the points raised during the review process.

We look forward to receiving your revised manuscript.

Kind regards,

Longxiu Huang, Ph.D.

Academic Editor

PLOS ONE

Journal Requirements:

Reviewers' comments:

Reviewer's Responses to Questions

**Comments to the Author**

1. If the authors have adequately addressed your comments raised in a previous round of review and you feel that this manuscript is now acceptable for publication, you may indicate that here to bypass the “Comments to the Author” section, enter your conflict of interest statement in the “Confidential to Editor” section, and submit your "Accept" recommendation.

Reviewer #1: All comments have been addressed

Reviewer #2: All comments have been addressed

Reviewer #3: All comments have been addressed

2. Is the manuscript technically sound, and do the data support the conclusions?

Reviewer #1: Yes

Reviewer #2: Yes

Reviewer #3: Partly

3. Has the statistical analysis been performed appropriately and rigorously? 

Reviewer #1: N/A

Reviewer #2: Yes

Reviewer #3: Yes

4. Have the authors made all data underlying the findings in their manuscript fully available?

Reviewer #1: Yes

Reviewer #2: Yes

Reviewer #3: Yes

5. Is the manuscript presented in an intelligible fashion and written in standard English?

Reviewer #1: Yes

Reviewer #2: Yes

Reviewer #3: Yes

6. Review Comments to the Author

Reviewer #1: Thanks for revising the manuscript.

Before it is published, I have three tiny comments:

1. A period is needed at the end of equations between the lines, especially equations in lines 241 and 249. For equations in lines 128 and 264, it would be better to add a comma at the end of the equations. For equation (1), I also suggest adding a comma or a semicolon at the end of the first line and a period at the end of the second line.

2. When mentioning "Supporting information," consider using "supporting information" or "Supporting Information." Also, since S1_File.pdf is the code, why not upload the raw code file directly?

3. The images in Figure 2 are still unclear. Although they might meet the PLOS requirements, I suggest using vector format for all images in the manuscript. If you prefer to use the current images, please enlarge the images in Figure 2 as they are hard to see carefully because they are small and some lines are close.

Reviewer #2: The authors are thanked for responding judiciously to my previous comments. By extending the experiments, and incorporating relevant changes in the revised paper, the authors have acted on both my recommendations and strengthened this manuscript considerably. One area of improvement is informing readers of the strengths and limitations of the proposed methodology.

In relation to changes in performance as the number of observations (N) is varied, the authors demonstrated that Type I error and FDR are consistently maintained at levels below 0.01; however statistical power is noticeably reduced when N is roughly below 40 (see Supporting Info S3). In relation to the significance threshold (Q), the author discovered that Q may be pushed from 0.01 to 0.05 to increase the True Positive Rate with negligible impact on the False Positive Rate, in the context of their experiments (see L425-444 and ROC curves in Fig.4). These additional insights elevate the quality of this contribution. Such guidance is invaluable as it assists researchers/practitioners in making informed choices during experimental design. The comments/discussions relating to non-Gaussian distributed features, convergence, cautionary remarks and techniques that specifically deal with nonlinearity (L197-205, L254-258 and L516-524, resp.) are appreciated.

The issues identified in the previous review have been satisfactorily addressed. On this basis, this reviewer recommends acceptance of this article for publication in PLOS.

A question is left below for the authors to ponder. This question (C-2) is, in my opinion, inconsequential for the publication decision. It is more about reflection and academic curiosity.

Minor comments

C-1 [Author comments, Figure R-1] The vertical axis appears to be mislabelled. Should it be the error of omission "1 - Power"?

C-2 [Author's response to Reviewer 2, Figure R-3] In the top-left plot for 2D simulated data, it is interesting to observe the Type 1 error increases with N. This trend is the opposite of FDR(N) which decreases as N increases.

a) This reviewer appreciates the Type 1 error is well controlled irrespective of N (remains below 0.01), but why are there fewer False Positives with fewer observations (for smaller N) and more False Positives with more observations (for larger N). If the Type I error is not under control, the inference would be "limit your observations" which seems a little counterintuitive.

b) If we consider a minimum volume ellipsoid (MVE) estimator, one would expect the location and scatter matrix (center and covariance structure of the ellipsoid) will be optimized to cover the inliers in the dataset, and the determinant of the positive definite symmetric matrix will be minimised in the process, subject to this cover constraint. In view of the finding in the Type 1 error graph, can we expect the volume to become more compact as an increasing function of N?

Reviewer #3: The authors have adequately addressed my comments raised in a previous round of review and I feel that the revised version is much better and can be accepted.

7. PLOS authors have the option to publish the peer review history of their article (what does this mean?). If published, this will include your full peer review and any attached files.

Reviewer #1: No

Reviewer #2: No

Reviewer #3: No

---

## [Decision Letter · Decision Letter 2]

29 Aug 2024

Hit screening with multivariate robust outlier detection

PONE-D-24-10015R2

Dear Dr. Leong,

We’re pleased to inform you that your manuscript has been judged scientifically suitable for publication and will be formally accepted for publication once it meets all outstanding technical requirements.

Kind regards,

Longxiu Huang, Ph.D.

Academic Editor

PLOS ONE

Reviewers' comments:

Reviewer's Responses to Questions

**Comments to the Author**

1. If the authors have adequately addressed your comments raised in a previous round of review and you feel that this manuscript is now acceptable for publication, you may indicate that here to bypass the “Comments to the Author” section, enter your conflict of interest statement in the “Confidential to Editor” section, and submit your "Accept" recommendation.

Reviewer #1: All comments have been addressed

2. Is the manuscript technically sound, and do the data support the conclusions?

Reviewer #1: Yes

3. Has the statistical analysis been performed appropriately and rigorously? 

Reviewer #1: Yes

4. Have the authors made all data underlying the findings in their manuscript fully available?

Reviewer #1: Yes

5. Is the manuscript presented in an intelligible fashion and written in standard English?

Reviewer #1: Yes

6. Review Comments to the Author

Reviewer #1: Thank you for addressing my comments and providing feedback on the image quality. Everything looks good for publication from my perspective. Best of luck with your future research!

7. PLOS authors have the option to publish the peer review history of their article (what does this mean?). If published, this will include your full peer review and any attached files.

Reviewer #1: No

---

## [Editor Report · Acceptance letter]

2 Sep 2024

PONE-D-24-10015R2 

PLOS ONE

Dear Dr. Leong, 

I'm pleased to inform you that your manuscript has been deemed suitable for publication in PLOS ONE. Congratulations! Your manuscript is now being handed over to our production team.

Kind regards, 

on behalf of

Dr. Longxiu Huang 

Academic Editor

PLOS ONE